# Kinetics of the Cell Cycle Arrest Biomarkers (TIMP2 and IGFBP7) for the Diagnosis of Acute Kidney Injury in Critically Ill COVID-19 Patients

**DOI:** 10.3390/diagnostics13020317

**Published:** 2023-01-15

**Authors:** Massimiliano Greco, Silvia De Rosa, Fabian Boehm, Sofia Spano, Romina Aceto, Antonio Voza, Francesco Reggiani, Marta Calatroni, Gianluca Castellani, Elena Costantini, Gianluca Villa, Maurizio Cecconi

**Affiliations:** 1Department of Biomedical Sciences, Humanitas University, Via Rita Levi Montalcini 4, Pieve Emanuele, 20072 Milan, Italy; 2Department of Anesthesia and Intensive Care, IRCCS Humanitas Research Hospital, Via Manzoni 56, Rozzano, 20089 Milan, Italy; 3Centre for Medical Sciences—CISMed, University of Trento, Via S. Maria Maddalena 1, 38122 Trento, Italy; 4Anesthesia and Intensive Care, Santa Chiara Regional Hospital, APSS Trento, 38122 Trento, Italy; 5Emergency Department, IRCCS Humanitas Research Hospital, Via Manzoni 56, Rozzano, 20089 Milan, Italy; 6Department of Nephrology, IRCCS Humanitas Research Hospital, Via Manzoni 56, Rozzano, 20089 Milan, Italy; 7Department of Health Sciences, Section of Anesthesia, Intensive Care and Pain Medicine, University of Florence, 50121 Florence, Italy; 8Department of Anesthesia and Intensive Care, Section of Oncological Anestehsia and Intensive Care, AOU Careggi, 50134 Florence, Italy

**Keywords:** acute kidney injury, critically ill, COVID-19, ARDS, [TIMP2]*[IGFBP7], renal replacement therapy, SARS-CoV-2

## Abstract

Background: Acute kidney injury (AKI) is highly prevalent in critical COVID-19 patients. The diagnosis and staging of AKI are based on serum creatinine (sCr) and urinary output criteria, with limitations in the functional markers. New cell-cycle arrest biomarkers [TIMP2]*[IGFBP7] have been proposed for early detection of AKI, but their role in critically ill COVID-19 patients is poorly understood. Methods: We conducted an observational study to assess the performance of [TIMP2]*[IGFBP7] for the detection of AKI in critical COVID-19 patients admitted to our intensive care unit (ICU). We sampled urinary [TIMP2]*[IGFBP7] levels at ICU admission, 12 h, 24 h, and 48 h, and compared the results to the development of AKI, as well as baseline and laboratory data. Results: Forty-one patients were enrolled. The median age was 66 years [57–72] and most were males (85%). Thirteen patients (31.7%) developed no/mild stage AKI, 19 patients (46.3%) moderate AKI, and nine patients (22.0%) severe AKI. The ICU mortality was 29.3%. sCr levels in the Emergency Department or at ICU admission were not significantly different according to AKI stage. [TIMP-2]*[IGFBP-7] urinary levels were elevated in severe AKI at 12 h after ICU admission, but not at ICU admission or 24 h or 48 h after ICU admission. Conclusion: Urinary biomarkers [TIMP-2]*[IGFBP-7] were generally increased in this population with a high prevalence of AKI, and were higher in patients with severe AKI measured at 12 h from ICU admission. Further studies are needed to evaluate the best timing of these biomarkers in this population.

## 1. Introduction

The COVID-19 pandemic, which began in December 2019, has been a burden on hospitals and intensive care units (ICU) across the globe, causing more than 6 million deaths [1]. While over 80% of cases may be mild, up to 5% of patients develop critical disease characterized by acute respiratory distress syndrome (ARDS), sepsis, and septic shock, with the need for mechanical ventilation [2]. Kidney involvement is common, and a meta-analysis including over 21,000 patients from three continents estimated that acute kidney injury (AKI) occurs in 39% of critical COVID-19 cases [3].

AKI, defined by the Kidney Disease: Improving Global Outcomes (KDIGO) workgroup, remains a challenge for hospitals and healthcare systems, with an estimated global incidence of 21.6% [4,5]. When considering only patients in the ICU, the incidence rises to a staggering 57.3%, with mortality rates of up to 23.9% [4,6]. Despite its prevalence, a report from the United Kingdom reported unacceptable delays in the recognition of hospital acquired AKI in 43% of patients [7]. Unfortunately, detecting AKI while in the subclinical or developing stage is a challenge.

Tissue inhibitor of metalloproteinase 2 (TIMP2) and insulin-like growth factor-binding protein 7 (IGFBP7) are two proteins that are involved in cell cycle arrest during the G1 phase, which occurs if cellular DNA damage is detected. They are considered indicators of cellular stress and are expressed throughout the body—notably also in the tubular cells of the kidneys [8,9,10]. Crucially, urinary concentrations of TIMP2 and IGFBP7 were found to have peaked 18 h after the initial decline in glomerular filtration rate (GFR), responding much faster than serum creatinine (sCr) [8]. TIMP2 and IGFBP7 are now considered reliable biomarkers for AKI prediction, diagnosis, and risk stratification [11].

While research on the applications of TIMP2 and IGFBP7 is promising in some settings [12,13,14], it is scarce in the context of COVID-19, with few and contradictory results [15,16]. The objective of this study was to assess the utility of [TIMP2]*[IGFBP7] biomarkers as predictors of severe AKI in a population of critically ill COVID-19 patients.

## 2. Materials and Methods

### 2.1. Study Design

This observational study included 41 patients who were admitted to the ICU at Humanitas Research Hospital in Rozzano, Italy from 30 March 2020 to 23 November 2020. Inclusion criteria were age >18 years, molecular confirmed Severe Acute Respiratory Syndrome Coronavirus 2 (SARS-CoV-2) infection, critical COVID-19 disease presenting with bilateral pneumonia, and admission to the ICU. Exclusion criteria were a negative SARS-CoV-2 PCR test, incomplete data for enrolment, and consent refusal. All consecutive patients enrolled underwent urine sampling for TIMP2 and IFGBP7 in a quality improvement process on the use of these biomarkers, which conformed to the Declaration of Helsinki. The study received IRB approval (no. 2485/2020). Informed consent for patient data collection and use was routinely obtained from all patients according to local procedures for critically ill patients. Study was registered in clinicaltrials (NCT04552340).

### 2.2. Definitions

AKI was defined and staged according to the KDIGO 2012 consensus guidelines [17,18], using both the sCr and urinary criteria, for each day for every patient included since hospital admission. AKI stage 1 was defined as an absolute increase in sCr of 0.3 mg/dL, a percentage increase in sCr of 50% (1.5-fold from baseline), or a reduction in urinary output (documented oliguria of <0.5 mL/kg per hour for more than 6 hours) [19]. AKI stage 2 was defined as a 2.0- to 2.9-fold increase in sCr from baseline, and AKI stage was defined as a 3- to 3.0-fold increase in sCr from baseline, an absolute increase in sCr of 4.0 mg/dL, or any AKI treated with renal replacement therapy (RRT) [19].

We considered AKI that developed within 72 h before ICU admission or up to 7 days thereafter as AKI related with ICU admission. The NephroCheck^®^ test, measuring urinary TIMP2 and IGFBP7, was performed. A [TIMP2]*[IGFBP7] value > 0.3 (ng/mL) 2/1000 was considered positive, while a value ≤ 0.3 was considered negative [20]. The final test output, labeled ‘AKI Risk’, is shown as a numeric score and was considered a continuous variable in this study.

### 2.3. Study Endpoints

The primary outcome was to assess the utility of [TIMP2]*[IGFBP7] biomarker as predictors of severe AKI in a population of critically ill COVID-19 patients.

Secondary outcomes were to analyze the development of AKI, need for continuous renal replacement therapy (CRRT), hospital and ICU length of stay, and survival status.

### 2.4. Data Collection

Detailed data were retrieved from ICU medical records. This included patients’ demographics, anthropometry, comorbidities, age, sex, weight (kg), duration of hospitalization, duration of ICU stay, blood urea nitrogen (mg/dL), IL-6 (pg/mL), hemoglobin (g/dL), ferritin, white blood cells, creatine kinase (U/L), maximum diuretic dosage prior to developing AKI (mg/day), urinary output (mL), sCr values (mg/dL), and fluid balance (mL/24 h). Data from local electronic health records were extracted and included in a separate database for data analysis. sCr was measured using the enzymatic method with an automatic analyzer (Dimension Vista, sCr Siemens Healthcare, Tarrytown, NY, USA). The concentration of biomarker [TIMP2]*[IGFBP7] was analyzed with the Astute 140 Meter Platform, using NephroCheck^®^ kits (bioMérieux S.A.-69280 Marcy l’Etoile-France). Urine samples for biomarker testing were collected sequentially at four time intervals: immediately upon ICU admission and 12, 24, and 48 h after. Urine testing for the [TIMP-2]*[IGFBP-7] concentration product yielded an absolute number known as the AKI Risk score, and was assessed by the Astute140 Meter [21]. Data on CRRT, length of ICU and hospital stay, hospital and ICU mortality, and dialysis discharge were also recorded.

### 2.5. Statistical Analyses

Values were expressed as frequencies and median interquartile ranges (IQR) and categorical variables as frequencies and percentages. The Mann-Whitney U test, Kruskal-Wallis test, and Chi-Square test were used, as appropriate, to compare variables between subgroups. The area under the curve (AUC) for the receiver operator characteristics curve was calculated for [TIMP-2]*[IGFBP7] and creatinine discriminative performance. All analyses were conducted in R 4.1.3 (a language and environment for statistical ## computing. R Foundation for Statistical Computing, Vienna, Austria).

## 3. Results

### Patients Characteristics

During the study period, 45 patients underwent screening. Four patients were excluded: two for invalid values and two for sampling/sample storage issues. Among the 41 patients eligible for analysis were 35 males (85%) and six females (15%) with a median age of 66 years. The majority of patients (68.3%) developed moderate to severe AKI, with only two patients not developing AKI of any degree. A total of nine patients (22%) progressed to severe AKI. RRT was initiated in two patients (5%).

Overall mortality was 29%, which was not correlated with AKI severity. Table 1 reports the baseline characteristics and outcomes according to AKI stage.

Among patients with severe AKI, the AKI risk score at 12 h from admission was according to the AKI stage at ICU admission, although it did not differ at 24 h or at 48 h, as reported in Figure 1.

Table 2 reports the laboratory values of sCr, urea, interleukin-6, hemoglobin, and creatinine-kinase at different time stages. The investigated laboratory markers did not differ according to the severity of AKI.

IL-6 values in the emergency room or upon ICU admission and at the maximum value are reported in Figure 2. They varied significantly over time but not according to stage of AKI. There was a low correlation between IL-6 at ICU admission and AKI risk, and a low correlation between the maximum value of IL-6 and AKI risk (R < 0.5, Appendix A).

Appendix A reports the means and standard errors for creatinine, IL-6, and AKI risk scores at different timepoints.

Discrimination analysis is reported in Figure 3 where we report the area under the curve of receiver operator characteristics curves for the detection of severe AKI with [TIMP-2]*[IGFBP7] at 12 h. AKI risk reached good discriminative performance (AUC = 0.810). 

Discrimination analysis of AKI risk at 12 h vs. sCr at ICU admission, and a combination of both markers is reported in Appendix A.

## 4. Discussion

In the present study, we demonstrated that urinary biomarkers [TIMP-2]*[IGFBP-7] were generally elevated at ICU admission and within the first 48 h in a population of critically ill COVID-19 patients with a high prevalence of AKI. In particular, the value of the biomarker was found to be increased only at 12 h in cases of more severe AKI.

The [TIMP-2]*[IGFBP-7] urinary biomarkers at 12 h from ICU admission had good performance for detecting severe AKI, compared to sCr. 

In this cohort, moderate to severe AKI developed in 68% of patients with critical COVID-19. This is significantly higher than the general incidence of AKI in the ICU, estimated at 39%. Nonetheless, this incidence was similar to the 76% AKI incidence reported in a study from New York during the first wave [3,22]. Differences in the severity of the population of critically ill patients and in the proportions of mechanically ventilated patients may account for discrepancies in the incidence of AKI. Our ICU had a prevalence of mechanically ventilated patients over 90% during the first and second wave, and all the patients included in our study underwent mechanical ventilation. This is consistent with previous data indicating a high prevalence of invasive ventilation in critical covid patients in the Lombardy region during the first months of the pandemic [23]. The high incidence of AKI in COVID-19 patients under mechanical ventilation may be due to dehydration related with several days of pneumonia including high fever, or with higher levels of positive end expiratory pressure (PEEP) initially used in severe C-ARDS patients, [24] as reported in several countries [25].

Acute kidney injury induced by COVID-19 has been recognized as a prominent source of morbidity and mortality. COVID-19-associated AKI has been linked to negative outcomes such as the development or worsening of comorbid diseases, increased mortality, and increased use of health care resources, according to previous research [26].

Moreover, the identification of AKI using existing criteria, which are based on an increase in sCr or a reduction in urine output, has some limitations. Consequently, several studies assessed the role of biomarkers for early AKI detection in COVID patients, with contrasting results. Unfortunately, in the human kidney, dysfunction is only visible when more than half of the renal mass is impaired and tubular damage markers can be used to detect AKI before filtration function is lost (subclinical AKI). For this purpose, we assessed [TIMP2]*[IGFBP7] biomarker for clinical use in our center.

A study on critically ill patients assessed the role of Neutrophil Gelatinase-Associated Lipocalin (NGAL) and [TIMP-2]*[IGFBP7] at ICU admission, and found the latter to be a risk factor for subsequent AKI development, even if AKI was less common in this population than in our population [15]. NGAL was not associated with AKI development, even if time to AKI was shorter in patients with increased NGAL levels. This is in contrast with results from other reports, showing an association between urinary NGAL levels and AKI development in COVID-19 patients [27,28]. Similarly, our data are in contrast to the findings of Husain-Syed et al., who found no correlation between [TIMP2]*[IGFBP7] and AKI in critical COVID-19 patients [16]. However, the population proposed is different from the one enrolled in our study, as only 12 patients were admitted to the ICU and nine patients were on mechanical ventilation, compared to our population of critically ill patients on invasive mechanical ventilation. Moreover, despite the standard proposed cut-off value of 0.3 to detect AKI, the majority of patients had higher levels of the AKI risk score at ICU admission in our study, with a median score of 0.64.

Regarding secondary outcomes, there was no difference in mortality according to AKI stage, in contrast with results from previous reports. The mortality rate of 29% is lower than the rates reported for New York City (50%) and other centers in Europe (62%) during the same period [22]. Multiple factors may have affected the association between AKI and mortality. High levels of PEEP and mean airway pressure during mechanical ventilation of severe ARDS may reduce renal function and precipitate AKI in patients that present with the most severe pulmonary disease [29]. Moreover, the use of diuretics and indications for CRRT therapy vs. high-dose diuretics for severe stages may differ between elderly or frail patients and younger and healthier patients, a known limit of AKI diagnostic and grading criteria [30]. The need for CRRT was low in this population, with only two out of 41 patients undergoing CRRT treatment [25]. Both the length of ICU and hospital stay showed a trend increase in patients with moderate and severe AKI, while the overall durations were similar to previously published reports [23].

The performance of sCr in predicting AKI at ICU admission was poor. The majority of patients had low creatinine levels upon admission to the ICU, and the median sCr measured at the time of ICU admission was the same for patients who developed moderate and severe AKI and those who did not (69.8 um/L for moderate AKI vs. 53.1 m/L for mild/no AKI). Even among patients with moderate and severe AKI, the median creatinine concentration was 124.7 um/L These findings do not establish a correlation between sCr and the risk of developing severe AKI. Even if a small trend in increasing creatinine levels across AKI severity classes could be detected, this trend was subclinical and not relevant to clinical practice.

We employed sequential urinary sampling to determine the optimal time window for using the urinary biomarkers to detect the onset of AKI. The AKI risk score at the time of ICU admission did not accurately reflect the development or severity of AKI. This could be due to a number of factors, including the fact that some patients were transported by other hospitals and that ICU admission was related with a rapidly worsening clinical picture and need for mechanical ventilation; consequently, the lack of a difference may be attributable to both patients and environmental factors. A number of patients were intubated upon admission to the ICU, and urinary specimens at ICU admission may not have yet reflected the damage caused by high PEEP levels and mechanical ventilation or need for prone position. As the [TIMP2]*[IGFBP-7] biomarker require a few hours to be detected in the urinary sample (4 h to 12 h), and ICU admission with mechanical ventilation was frequently the culmination of a rapid deterioration from the ward with precipitating respiratory failure, the urinary specimen at 12 h, which were increased in this population, may be the optimal time-window to detect the clinical deterioration and early kidney damage in our population. Nonetheless, the small number of patients and population variability of this study may limit the interpretability of analyses at different time points and should be taken only as hypothesis generating. Larger sample size studies are required to reach a definitive conclusion. This is also suggested by the fact that urinary sampling at later stages, 24 h and 48 h, did not reflect severity of kidney damage. The progression of urinary [TIMP2]*[IGFBP7] levels from mild to moderate to severe AKI enhances the biological plausibility of the 12-h window.

The advantages of biological markers of AKI in COVID-19 patients could be the same as in other populations, to overcome the limitations of a standard functional biomarker, even if cut-off values for decision making need further studies. Accordingly, the current KDIGO guidelines [5] are hindered by a number of factors that biological markers could address. First, because the KDIGO criteria of sCr and urine output can be used independently of one another, many studies omit urine output as a diagnostic criterion of AKI, thereby artificially reducing the incidence of AKI, while biomarkers could help in standardizing incidence reports. In addition, the urinary output criterion is frequently dependent on the choice of diuretic administration, which may lead to erroneous conclusions regarding urinary volume [31], and its specificity and sensitivity are often a matter of debate. Similarly, sCr’s limitations in the detection and evaluation of AKI have been well known and documented for years [32]. Creatinine is essentially used as a surrogate for the glomerular filtration rate GFR, as a decreased GFR typically results in elevated sCr levels after several hours, and fluctuations in sCr lag several days behind the actual onset of AKI. As is well known from published studies, several factors may affect sCr levels in critical patients [33,34], hindering the clinical application of this biomarker. In addition, sCr is incapable of detecting subclinical AKI, resulting in a delayed diagnosis. Urinary biomarker [TIMP2]*[IGFBP7] have a biological advantage over sCr due to their expression in kidney tubule cells, where they serve as stress indicators of tubular cells. Although the progression does not always occur, tubular cellular stress is often an early sign of kidney damage [10]. Therefore, TIMP2 and IGFBP7 can detect AKI while it is still in an early or subclinical stage, allowing for early preventative treatment. The findings of this study suggest that using [TIMP2]*[IGFBP7] at 12 h could lead to the early detection of severe AKI, prompting additional clinical action to hinder further kidney damage.

### Limitations

This is an observational study including a small population of 41 critically ill patients. Although the population size is small, this is comparable to size of other studies on the use of biomarkers to detect AKI in COVID-19 patients [15,16]. Compared to multi-centric studies, a single-center study may hinder the generalizability of its findings. On the other hand, the findings in this population may reflect the unique characteristics of the first and second covid wave of critically ill patients, which may reveal the high workload and strain on hospital and intensive care units due to the high number of hospitalized COVID-19 patients in a limited time span.

Due to time constraints during the pandemic and the emergency setting, some useful clinical data could not be collected, including data on therapies, and all possible laboratory markers. Moreover, we were not able to collect all patient data for the whole length of ICU admission. In the absence of baseline sCr values for all patients, the lowest values during hospital admission were used as a reference to calculate AKI and AKI stage, as advocated by the KDIGO guidelines. While this is a method suggested by the guidelines, it may increase the incidence of AKI in the study population.

## 5. Conclusions

In critical COVID-19 patients, the urinary biomarker [TIMP2]*[IGFBP7] were generally elevated, and were higher in severe AKI when sampled at 12 h after ICU admission, but not at other timepoints. These findings are based on a limited population and should be confirmed by larger studies. Further research is required to determine the clinical utility and optimal cut-off values of these markers in critically ill Covid-19 patients.

## Figures and Tables

**Figure 1 diagnostics-13-00317-f001:**
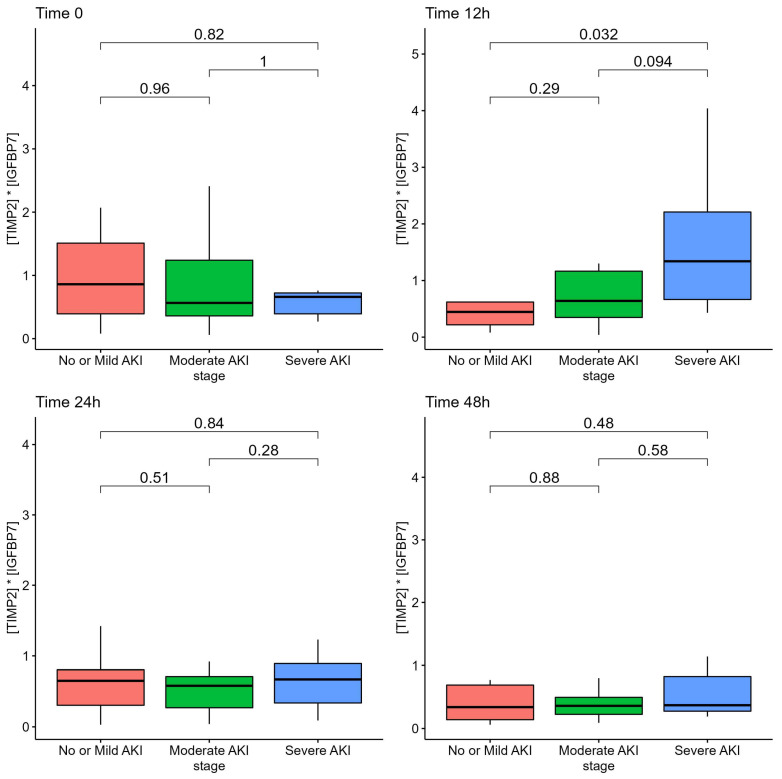
AKI Risk Score ([TIMP2]*[IGFBP7] biomarkers) of patients according to AKI stage.

**Figure 2 diagnostics-13-00317-f002:**
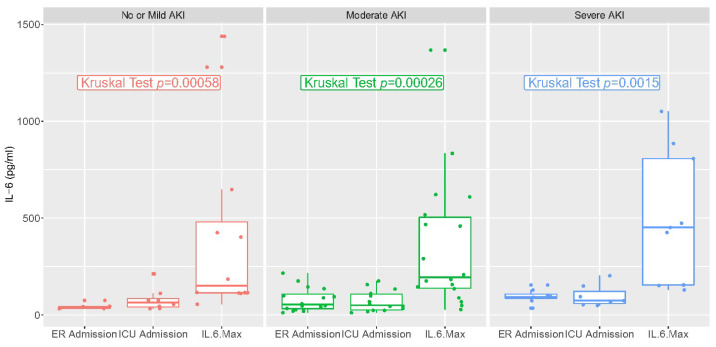
IL-6 values at different time points: Emergency Department admission, ICU admission, and highest levels during hospital stay.

**Figure 3 diagnostics-13-00317-f003:**
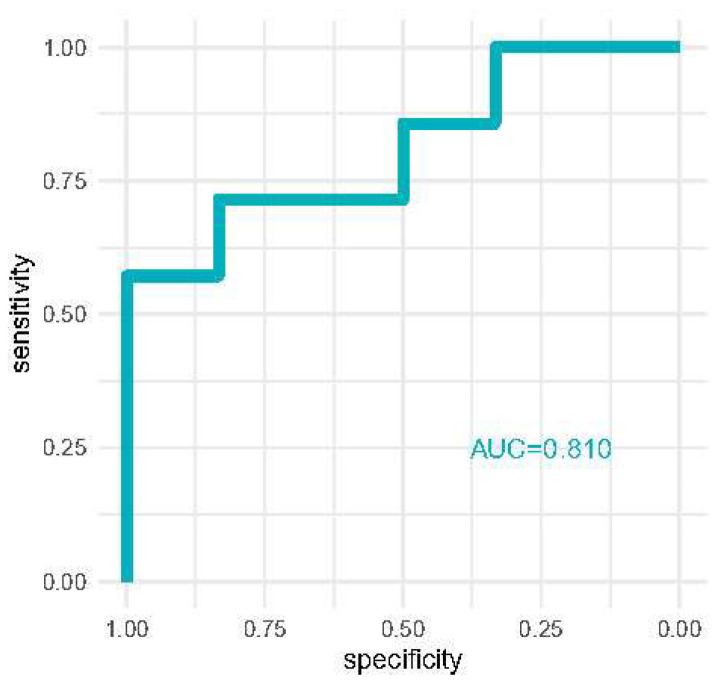
ROC curve comparing the AKI risk score for diagnosing severe AKI compared to milder AKI stages.

**Table 1 diagnostics-13-00317-t001:** Baseline and outcome data according to AKI stage.

Characteristic	N	Overall, *n* = 41 ^1^	No/Mild AKI, *n =* 13 ^1^	Moderate AKI, *n =* 19 ^1^	Severe AKI, *n =* 9 ^1^	*p*-Value ^2^
Age at Admission	41	66 (57, 72)	66 (57, 67)	63 (56, 73)	71 (59, 72)	0.5
Male gender	41	35.0 (85.4%)	12.0 (92.3%)	15.0 (78.9%)	8.0 (88.9%)	0.7
Hypertension	41	16 (39.0%)	3 (23.1%)	8 (42.1%)	5 (55.6%)	0.3
Diabetes Mellitus	41	8.0 (19.5%)	2 (15.4%)	3 (15.8%)	3 (33.3%)	0.6
Moderate/Severe Obesity	41	4.0 (9.8%)	2 (15.4%)	2 (10.5%)	0 (0%)	0.3
Coronary artery disease	41	6 (14.6%)	2 (15.4%)	4.0 (21.1%)	0 (0%)	0.4
No previous pathology	41	12 (29.3%)	4.0 (30.8%)	6.0 (31.6%)	2.0 (22.2%)	0.9
Length of symptoms before hospital admission (days)	39	6.00 (3.50, 7.00)	5.50 (3.75, 7.00)	6.50 (3.25, 7.75)	5.00 (5.00, 7.00)	0.8
Length of ICU stay	41	15 (10, 24)	12 (11, 14)	15 (8, 22)	41 (31, 46)	0.005
Length of Hospital stay	41	27 (19, 41)	24 (19, 29)	22 (17, 30)	63 (56, 70)	0.003
ICU Mortality	41	12.0 (29.3%)	4.0 (30.8%)	8.0 (42.1%)	0.0 (0.0%)	0.062
Hospital Mortality	41	12.0 (29.3%)	4.0 (30.8%)	8.0 (42.1%)	0.0 (0.0%)	0.062
Need for CRRT	41	2.0 (4.9%)	0.0 (0.0%)	0.0 (0.0%)	2.0 (22.2%)	0.044
Dialysis at discharge	41	0.0 (0.0%)	0.0 (0.0%)	0.0 (0.0%)	0.0 (0.0%)	

^1^ Median (IQR) or Frequency (%), ^2^ Kruskal-Wallis test; Fisher’s exact test. Abbreviations: CRRT, Continuous Renal Replacement Therapy; ICU, Intensive Care Unit.

**Table 2 diagnostics-13-00317-t002:** Laboratory data at different time points according to AKI stage.

Characteristic	N	Overall, *n =* 41 ^1^	No/Mild AKI, *n =* 13 ^1^	Moderate AKI, *n =* 19 ^1^	Severe AKI, *n =* 9 ^1^	*p*-Value ^2^
Emergency Department						
sCr at ER admission (μmol/L)	36	88 (69, 113)	82 (55, 115)	87 (70, 115)	108 (90, 111)	0.5
Urea at ER (mmol/L)	36	7.69 (5.99, 9.16)	7.08 (5.97, 9.03)	7.24 (6.01, 8.56)	9.11 (8.11, 9.52)	0.4
IL-6 at ER admission (pg/mL)	30	64 (35, 97)	40 (34, 44)	52 (31, 107)	92 (86, 107)	0.10
WBC (10^3^/mm^3^)	38	7.0 (5.2, 10.1)	8.6 (5.1, 12.2)	6.8 (5.1, 8.4)	6.7 (6.0, 8.9)	0.7
Hemoglobin at ER admission (g/dL)	37	14.50 (13.20, 15.30)	13.50 (13.05, 14.20)	14.85 (13.17, 15.28)	15.30 (14.07, 16.23)	0.11
ICU admission						
sCr at ICU admission (μmol/L)	41	70 (57, 95)	62 (54, 95)	70 (59, 84)	83 (69, 110)	0.3
Urea at ICU admission (mmol/L)	41	7.48 (6.01, 9.96)	6.73 (6.58, 8.33)	7.66 (5.94, 10.41)	8.24 (5.89, 10.94)	0.6
IL-6 at ICU admission (pg/mL)	29	65 (44, 110)	64 (41, 85)	50 (26, 107)	73 (59, 122)	0.4
Ferritin (ng/mL)	40	1256 (526, 1540)	852 (524, 1376)	1352 (320, 1674)	1422 (1119, 1882)	
WBC (10^3^/mm^3^)	41	9.8 (7.2, 12.3)	10.5 (7.1, 13.8)	8.5 (6.9, 10.5)	11.8 (9.5, 12.3)	0.3
Hemoglobin at ICU admission (g/dL)	41	13.20 (12.00, 14.60)	13.20 (12.90, 14.30)	13.20 (12.20, 14.90)	13.50 (11.80, 14.60)	>0.9
ICU Highest value						
sCr Maximum value (μmol/L)	41	101 (75, 127)	92 (70, 119)	103 (79, 126)	125 (95, 323)	0.2
Urea Maximum value (mmol/L)	41	14 (11, 21)	14 (9, 21)	14 (11, 19)	18 (14, 31)	0.11
IL-6 Maximum value (pg/mL)	39	207 (122, 564)	150 (114, 480)	195 (137, 504)	451 (154, 807)	0.4
WBC (10^3^/mm^3^)	41	18 (12, 22)	18 (14, 18)	15 (10, 21)	22 (19, 31)	0.034
Creatine Kinase Maximum Value	41	255 (149, 448)	236 (149, 731)	246 (156, 437)	316 (225, 448)	0.9
Discharge						
Urea at ICU Discharge mmol/L)	41	8.4 (6.6, 14.2)	10.9 (5.3, 15.8)	8.4 (7.4, 13.0)	8.1 (6.1, 12.8)	0.7
sCr at ICU Discharge ICU (μmol/L)	41	57 (43, 68)	65 (43, 79)	56 (48, 65)	57 (38, 129)	0.8
Urea at Hospital Discharge (mg/dL)	41	6 (3, 11)	5 (3, 10)	7 (4, 13)	4 (3, 6)	0.2
sCr at Hospital Discharge (mmol/L)	41	59 (46, 70)	59 (52, 68)	58 (50, 65)	68 (40, 75)	>0.9

^1^ Median (IQR) or Frequency (%), ^2^ Kruskal-Wallis test. Abbreviations: ICU, Intensive Care Unit; ER: Emergency Room; IL-6, Interleukin 6; sCr, Serum Creatinine.

## Data Availability

Data available from the authors upon reasonable request, and after internal hospital revision for data sharing by the hospital research hoffice.

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
