# Peer review of "Kinetics of the Cell Cycle Arrest Biomarkers (TIMP2 and IGFBP7) for the Diagnosis of Acute Kidney Injury in Critically Ill COVID-19 Patients"

_diagnostics, 2023, doi:10.3390/diagnostics13020317_

Round 1
Reviewer 1 Report
The authors evaluate the utility of a promising biomarker of early kidney damage (TIMP2*IGFBP7) to detect AKI in critically ill COVID-19 patients.
The study is well planned and the possible clinical utility of this biomarker is evident, due to limitations of classical biomarkers and the high incidence of AKI in patients who suffer severe clinical effects after infection with SARS-CoV-2.
However, the authors should review/clarify some points before the publication of their article:
-In relation to the primary outcome (lines 99-100): "TIMP2 and IGFBP7" should be replaced by "TIMP2 *IGFBP7", since the product of both ("Nephrocheck") is really evaluated and not the biomarkers independently.
-The secondary outcomes (lines 101 and 102) are barely discussed. If they are an study objective, obtained results should be conveniently discussed.
-The units of TIMP2*IGFBP7 must be clarified. Why divide the result by 1000? (line 94). If the results are already in ng/mL, the explanation on lines 114-115 does not apply.
-Lines 117-121: AKI risk Score is synonymous with TIMP2 *IGFBP7? This concept should be clarified in the text. On the other hand, the use of the Astute 140 Meter is repeated throughout the paragraph: please unify.
-Why did “No” and “Mild AKI” join? Wouldn't it have been better to separate them and work with 4 groups?
-Table 2: although there are no significant differences, a trend of increased plasma creatinine and urea was observed in groups with Severe AKI. Maybe differences are not significant due to the low N. If this trend is confirmed, there would already be differences on admission time for these markers... So, what would TIMP2*IGFBP7 add to AKI diagnosis?
- Figure 2 repeats information from Figure 1. In my opinion it can be eliminated. In any case, it should be placed after Figure 1 and groups should be ordered: “No/Mild AKI” appears in the middle of the other two groups (when it should be the first).
-Figure 3: the purpose of measuring this marker should be explained as well as its results discussed. At what time does IL6 rise? 12, 24, 48 hours? Only the maximum peak result appears. On the other hand, authors should order the groups: “No/Mild” appears in the middle of the other two groups (when its should be first). The axes/legends of this Figure are barely visible.
-The figure 4 caption should be revised. Are the authors sure that they are comparing AKI Risk Score and sCr?? In my opinion, they are evaluating the predictive capacity of AKI Risk Score 12 hours after admission to the ICU.
-A very interesting point in the article results is the finding of elevated TIMP2*IGFBP7 values ​​in critically ill patients with COVID (with or without severe AKI). This point should be further discussed.
- TIMP2*IGFBP7 rises significantly 12 hours after admission to the ICU, but not at 0, 24, and 48 hours. The reasons given by the authors to explain this are not entirely convincing. …Could it be due to other causes? Perhaps N is too low to draw any conclusions?
Other minor comments:
- There is an author for whom there is no affiliation.
- Abbreviations should be checked. As an example: definition of "UCI" does not appear in the abstract; it sometimes appears without the abbreviation in the text (line 72).
- Nephrocheck is a registered trademark, please indicate it as such.
- Line 137: what does 3% indicate?
- Tables should indicate between which groups there are significant differences (if any).
- In Figures 1 and 2, AKI Risk Score units should be unified and clarified.
- Line 211: where it says "AKI at the time...", perhaps the authors mean "AKI Risk score at the time..."
- Line 253: what does “AKI.19” mean?
Author Response
Reviewer 1
The authors evaluate the utility of a promising biomarker of early kidney damage (TIMP2*IGFBP7) to detect AKI in critically ill COVID-19 patients.
The study is well planned and the possible clinical utility of this biomarker is evident, due to limitations of classical biomarkers and the high incidence of AKI in patients who suffer severe clinical effects after infection with SARS-CoV-2.
We thank the reviewer for this nice comment on our manuscript
However, the authors should review/clarify some points before the publication of their article:
-In relation to the primary outcome (lines 99-100): "TIMP2 and IGFBP7" should be replaced by "TIMP2 *IGFBP7", since the product of both ("Nephrocheck") is really evaluated and not the biomarkers independently.
Thank you, we have modified as suggested
-The secondary outcomes (lines 101 and 102) are barely discussed. If they are an study objective, obtained results should be conveniently discussed.
We now have increased the discussion of these issues in the discussion section, where we write.
“Regarding secondary outcomes, there was no difference in mortality according to AKI stage in our population, in contrast with results from previous reports. The mortality rate of 29% is lower than the rates reported for New York City (50%) and other centers in Eu-rope (62%) during the same period[22]. Multiple factors may have affected the association between AKI and mortality. High levels of PEEP and mean airway pressure during me-chanical ventilation of severe ARDS may reduce renal function and precipitate AKI in pa-tients which present the most severe pulmonary disease [29]. Moreover, the use of diuret-ics and indications for CRRT therapy vs high dose diuretics may differ between elderly and frail patients and younger patients and healthier patients, a known limit of AKI di-agnostic and grading criteria[30].
Need for CRRT was low in this population, with only 2 patients out of 41 undergoing CRRT treatment [25]. Both length of ICU and hospital stay showed a trend for increase in patients with moderate and severe AKI, while the overall duration were similar to previ-ous published reports [23].”
-The units of TIMP2*IGFBP7 must be clarified. Why divide the result by 1000? (line 94). If the results are already in ng/mL, the explanation on lines 114-115 does not apply.
We agree and we have now improved the text to clarify this, as suggested by the reviewer
-Lines 117-121: AKI risk Score is synonymous with TIMP2 *IGFBP7? This concept should be clarified in the text. On the other hand, the use of the Astute 140 Meter is repeated throughout the paragraph: please unify.
We agree, and now better specify as follows
“Urine testing for [TIMP-2]*[IGFBP-7] concentration product yielded an absolute number known as AKI Risk score, and were assessed by the Astute140 Meter (Astute Medical)”
-Why did “No” and “Mild AKI” join? Wouldn't it have been better to separate them and work with 4 groups?
We agree with the reviewer, but as reported at the beginning of the results, we only had 2 patients not developing AKI. In the first and second COVID-19 waves in Lombardy, most of the patients had at least some form of AKI as defined by KDIGO criteria, a percentage which was found to be lower in subsequent waves. Given the small numerosity of this group, we have joined low risk patient (no AKI or AKI stage 1) in the same low risk group.
-Table 2: although there are no significant differences, a trend of increased plasma creatinine and urea was observed in groups with Severe AKI. Maybe differences are not significant due to the low N. If this trend is confirmed, there would already be differences on admission time for these markers... So, what would TIMP2*IGFBP7 add to AKI diagnosis?
The reviewer is right in pointing out the trend in creatinine. However, the trend is evident only at population level, while it is very difficult to detect any trend for worsening function at clinical level, especially when baseline serum creatinine levels are not known by the caring physician. Below standard threshold for increased creatinine, is impossible for clinicians to assess the severity of kidney damage, also given the large reserve function of kidneys. In this setting, the use of biomarkers may be useful, to anticipate actions and avoid further kidney damage.
We now better discuss this in discussion, where we write
“Even if a small trend in increasing creatinine levels across AKI severity classes could be detected, this trend was subclinical and not relevant to clinical practice.”
- Figure 2 repeats information from Figure 1. In my opinion it can be eliminated. In any case, it should be placed after Figure 1 and groups should be ordered: “No/Mild AKI” appears in the middle of the other two groups (when it should be the first).
Thank you, we have now removed this figure, as suggested by the reviewer
-Figure 3: the purpose of measuring this marker should be explained as well as its results discussed. At what time does IL6 rise? 12, 24, 48 hours? Only the maximum peak result appears. On the other hand, authors should order the groups: “No/Mild” appears in the middle of the other two groups (when its should be first). The axes/legends of this Figure are barely visible.
We thank the reviewer for this suggestion. However, we only measured IL-6 in the Emergency Department (ER Admission), at ICU admission, and have the peak IL-6 levels of the ICU stay. We now better specify this in Figure legend. Thank you, we have now re-ordered the groups of this figure, and better rendered the axis.
-The figure 4 caption should be revised. Are the authors sure that they are comparing AKI Risk Score and sCr?? In my opinion, they are evaluating the predictive capacity of AKI Risk Score 12 hours after admission to the ICU.
We thank the reviewer for pointing this. This has now been corrected.
-A very interesting point in the article results is the finding of elevated TIMP2*IGFBP7 values ​​in critically ill patients with COVID (with or without severe AKI). This point should be further discussed.
We agree with the reviewer, and have now added a more extensive discussion of the use of biomarkers in COVID-19 patients. We now write in discussion:
“Acute kidney injury induced by COVID-19 has been recognized as a prominent source of morbidity and mortality. COVID-19- associated AKI (COVID-19 AKI) has been linked to negative outcomes such as the development or worsening of comorbid diseases, increased mortality, and increased use of health care resources, according to research. [26] Nevertheless, the identification of AKI based on existing criteria, which are based on an increase in serum creatinine or a reduction in urine output, has some limitations. Consequently, several studies assessed the role of biomarkers for early AKI detection in COVID patients, with contrasting results. A study on critically ill patients assessed the role of Neutrophil Gelatinase-Associated Lipocalin (NGAL) and [TIMP-2] × [IGFBP7] at admission in critical areas, and found the latter to be a risk factor for subsequent AKI development, even if AKI was less common in this population than in our population [. 15]. NGAL was not associated with AKI development, even if time to AKI was shorter in pa-tients with increased NGAL levels. This is in contrast with results from other reports, showing an association between urinary NGAL levels and AKI development in COVID-19 patients[27,28]. Similarly, our data are in contrast to the findings of Husain-Syed et al., who found no correlation between [TIMP2]*[IGFBP7] and AKI in critical COVID-19 patients[16]. However, the population proposed is different from the one proposed in our study, as only 12 patients admitted to ICU and 9 patients on mechanical ventilation, compared to our population of critically ill patients under mechanical ventilation. More-over, despite the standard proposed cut-off value of 0.3 to detect AKI, the majority of patients had higher levels of AKI risk score at ICU admission in our study, with a median population score of 0.64.”
- TIMP2*IGFBP7 rises significantly 12 hours after admission to the ICU, but not at 0, 24, and 48 hours. The reasons given by the authors to explain this are not entirely convincing. …Could it be due to other causes? Perhaps N is too low to draw any conclusions?
We agree with the reviewer, and have included the sample size in the limitation settings, and we now better discuss this in the conclusion and discussion settings, where we write
“In critical COVID-19 patients the urinary biomarkers [TIMP2]*[IGFBP7] were gener-ally elevated, and were higher in severe AKI when sampled at 12 h after ICU admission, but not at other timepoints. These findings are based on a limited population, and should be confirmed by larger studies. Further research is required to determine the clinical utility and optimal cut-off values of these markers in critically ill Covid-19 patients.”
Other minor comments:
- There is an author for whom there is no affiliation.
Thank you, we have corrected this
- Abbreviations should be checked. As an example: definition of "UCI" does not appear in the abstract; it sometimes appears without the abbreviation in the text (line 72).
Thank you, we have corrected this
- Nephrocheck is a registered trademark, please indicate it as such.
Thank you, we have corrected this
- Line 137: what does 3% indicate?
Thank you, we have now corrected this
- Tables should indicate between which groups there are significant differences (if any).
Tables report the results of Kruskal test across the 3 categories, we now specify this in table caption
- In Figures 1 and 2, AKI Risk Score units should be unified and clarified.
We have now removed figure 2, as suggested by the reviewer
- Line 211: where it says "AKI at the time...", perhaps the authors mean "AKI Risk score at the time..."
Yes, the reviewer is correct, and we have now corrected this inaccuracy
- Line 253: what does “AKI.19” mean?
Thank you, we have now corrected this
Reviewer 2 Report
In this original article, the authors have explored the kinetics of the cell cycle arrest biomarkers TIMP2 and IGFBP7 in critically ill COVID-19 patients who developed acute kidney injury.
The paper is overall well-written and the study seems scientifically sound.
Several suggestions for revision are listed below:
Minor corrections
1. There are minor spelling errors, e.g., the dot should be after the reference in brackets and not before, i.e., p5[5]. not .[5]
2. The references are not formatted according to the journal's guidelines- Diagnostics requires American Chemical Society citation style
Major corrections
1. The authors should also discuss other publications that assessed these markers in COVID-19. See the following MEDLINE articles and also search for abstracts presented at international conferences that might have investigated the topic
https://pubmed.ncbi.nlm.nih.gov/?term=%28%22TIMP2%22+OR+%22IGFBP7%22%29+AND+%22covid-19%22&show_snippets=off&sort=date
2. You did not present any data on the other comorbidities of these patients. Did they not have any associated disorders? Diabetes? Hypertension? Comorbidities should have been taken into consideration in your statistical analysis as well.
3. You only reported several laboratory data. How about WBC? Ferritin? Uric acid? See the following paper for other biomarkers altered in COVID-19: https://scholar.valpo.edu/jmms/vol9/iss1/5/ etc.
4. Did any of these patients associate other infections? UTIs or other infections that could have affected renal function?
5. How about the treatment received? Antibiotics? NSAIDs? Antiviral agents? Maybe AKI occurred also secondary to the medicines prescribed.
Overall, I think this paper has significant biases, requiring a major revision at this point.
Author Response
Reviewer 2
In this original article, the authors have explored the kinetics of the cell cycle arrest biomarkers TIMP2 and IGFBP7 in critically ill COVID-19 patients who developed acute kidney injury.
The paper is overall well-written and the study seems scientifically sound.
We are very grateful to the reviewer for this good feedback on our manuscript
Several suggestions for revision are listed below:
Minor corrections
- There are minor spelling errors, e.g., the dot should be after the reference in brackets and not before, i.e., p5[5]. not .[5]
We have now corrected this issue throughout, and all the dots are after the reference in brackets.
- The references are not formatted according to the journal's guidelines- Diagnostics requires American Chemical Society citation style
We have now corrected this issue, and reformatted references according to American Chemical Society style using mendely app
Major corrections
- The authors should also discuss other publications that assessed these markers in COVID-19. See the following MEDLINE articles and also search for abstracts presented at international conferences that might have investigated the topic
https://pubmed.ncbi.nlm.nih.gov/?term=%28%22TIMP2%22+OR+%22IGFBP7%22%29+AND+%22covid-19%22&show_snippets=off&sort=date
We agree with the reviewer, and now cite more papers regarding COVID-19 and urinary biomarkers in the discussion, where we now write
“Acute kidney injury induced by COVID-19 has been recognized as a prominent source of morbidity and mortality. COVID-19- associated AKI (COVID-19 AKI) has been linked to negative outcomes such as the development or worsening of comorbid diseases, increased mortality, and increased use of health care resources, according to research. [26] Nevertheless, the identification of AKI based on existing criteria, which are based on an increase in serum creatinine or a reduction in urine output, has some limitations. Consequently, several studies assessed the role of biomarkers for early AKI detection in COVID patients, with contrasting results. A study on critically ill patients assessed the role of Neutrophil Gelatinase-Associated Lipocalin (NGAL) and [TIMP-2] × [IGFBP7] at admission in critical areas, and found the latter to be a risk factor for subsequent AKI development, even if AKI was less common in this population than in our population [. 15]. NGAL was not associated with AKI development, even if time to AKI was shorter in pa-tients with increased NGAL levels. This is in contrast with results from other reports, showing an association between urinary NGAL levels and AKI development in COVID-19 patients[27,28]. Similarly, our data are in contrast to the findings of Husain-Syed et al., who found no correlation between [TIMP2]*[IGFBP7] and AKI in critical COVID-19 patients[16]. However, the population proposed is different from the one proposed in our study, as only 12 patients admitted to ICU and 9 patients on mechanical ventilation, compared to our population of critically ill patients under mechanical ventilation. More-over, despite the standard proposed cut-off value of 0.3 to detect AKI, the majority of patients had higher levels of AKI risk score at ICU admission in our study, with a median population score of 0.64.”
- You did not present any data on the other comorbidities of these patients. Did they not have any associated disorders? Diabetes? Hypertension? Comorbidities should have been taken into consideration in your statistical analysis as well.
We thank the reviewer for this helpful suggestion. We now add the data on patient comorbidities to table 1, as requested by the reviewer
- You only reported several laboratory data. How about WBC? Ferritin? Uric acid? See the following paper for other biomarkers altered in COVID-19: https://scholar.valpo.edu/jmms/vol9/iss1/5/ etc.
We thank the reviewer for this helpful suggestion. We now add data on WBC and Ferritin, as requested by the reviewer, while we didn’t collect data on Uric Acid in this population. Data are added to table 2
- Did any of these patients associate other infections? UTIs or other infections that could have affected renal function?
In all the patients included, respiratory failure related with COVID-19 ARDS was the primary reason for ICU admission. There was no urinary infection, and other co-infection at ICU admission were rare in the first and second wave, and often difficult to diagnose radiologically and from available biomarkers. We did not collect data on co-infection during ICU stay, but the time window of this study is limited to the first days after ICU admission.
- How about the treatment received? Antibiotics? NSAIDs? Antiviral agents? Maybe AKI occurred also secondary to the medicines prescribed.
We agree with the reviewer that some of these data could have been useful, but unfortunately some of these data could not be collected during the emergency phase of first and second covid wave in our hospital. Several of data on therapies were collected on paper records, which were used across 4 surge-ICU in emergency settings, and these data were not collected
Overall, I think this paper has significant biases, requiring a major revision at this point.
We have performed several major revisions to the manuscript and hope now this is in a better form compared to previous reports.
Reviewer 3 Report
Although this manuscript has some interesting points of view, I am not convinced about the authors’ conclusion regarding elevated [TIMP2]*[IGFBP7] at 12h in COVID-19 patients with severe AKI. Four statistical calculations were performed on AKI risk scores, but a Bonferroni correction was not applied. When such a correction is made, the significance is no longer at hand.
Husain-Syed et al (PMID: 32691068) found that [TIMP-2] • [IGFBP7] had no effect on predicting AKI in patients with COVID-19, but higher levels were associated with adverse clinical outcomes, including the severity of AKI.
Thus, the new discoveries added in the present manuscript are somewhat limited.
Irqsusi and co-workers (PMID: 35001430) noted a significant increase in [TIMP2]*[IGFBP7] at 24h, but not after 6h or 12h, respectively, in post-cardiac surgery patients. This increase was strengthened when [TIMP2]*[IGFBP7] was divided by osmolality.
Possibly, this could be an approach to strengthen the scientific value of this manuscript.
The authors have put a lot of effort in this paper, which, overall, is well-written, but I am sorry to say that I cannot approve its publication in its present form.
If the study has an NCT number, this should be shown.
Author Response
Reviewer 3
Although this manuscript has some interesting points of view, I am not convinced about the authors’ conclusion regarding elevated [TIMP2]*[IGFBP7] at 12h in COVID-19 patients with severe AKI. Four statistical calculations were performed on AKI risk scores, but a Bonferroni correction was not applied. When such a correction is made, the significance is no longer at hand.
We thanks to the reviewer for the feedback on our manuscript; the small sample size of the population and the high prevalence of AKI in this community of critically ill patients are an acknowledged limitation of this study, but at the same time this study as a similar sample size to the study published on other kidney injury biomarkers in COVID patients. We now have improved both the discussion, the limitation section and the conclusion of the study to better reflect the limitation of the available data. In the conclusion now we writ
“In critical COVID-19 patients the urinary biomarkers [TIMP2]*[IGFBP7] were generally elevated, and were higher in severe AKI when sampled at 12 h after ICU admission, but not at other timepoints. These findings are based on a limited population, and should be confirmed by larger studies. Further research is required to determine the clinical utility and optimal cut-off values of these markers in critically ill Covid-19 patients.”
Husain-Syed et al (PMID: 32691068) found that [TIMP-2] • [IGFBP7] had no effect on predicting AKI in patients with COVID-19, but higher levels were associated with adverse clinical outcomes, including the severity of AKI.
Thus, the new discoveries added in the present manuscript are somewhat limited.
We thank the reviewer for this advice. We were citing the study by Husain-Syed reference in our discussion (previous reference number 17). Thanks to the suggestion of the reviewer, we now better underline the difference between the population of study and our population, and now in discussion we write
“our data are in contrast to the findings of Husain-Syed et al., who found no correlation between [TIMP2]*[IGFBP7] and AKI in critical COVID-19 patients[16]. However, the pop-ulation proposed is different from the one proposed in our study, as only 12 patients ad-mitted to ICU and 9 patients on mechanical ventilation, compared to our population of critically ill patients under mechanical ventilation. Moreover, despite the standard pro-posed cut-off value of 0.3 to detect AKI, the majority of patients had higher levels of AKI risk score at ICU admission in our study, with a median population score of 0.64.”
Irqsusi and co-workers (PMID: 35001430) noted a significant increase in [TIMP2]*[IGFBP7] at 24h, but not after 6h or 12h, respectively, in post-cardiac surgery patients. This increase was strengthened when [TIMP2]*[IGFBP7] was divided by osmolality.
Possibly, this could be an approach to strengthen the scientific value of this manuscript.
We agree with the reviewer, unfortunately we don’t have urinary osmolarity, as the study was conducted in the first weeks of the first wave and second wave in a busy covid hospital in Lombardy, were teams were struggling to cope with waves of covid patients, and not all data could be collected. Urinary osmolarity was not collected in these patients in the first days of ICU admission.
The authors have put a lot of effort in this paper, which, overall, is well-written, but I am sorry to say that I cannot approve its publication in its present form.
We appreciate the reviewer's constructive feedback on our paper, which we think we have now significantly improved through his corrections and those of the other reviewers. Very few reports were published on the [TIMP2]*[IGFBP7] urinary markers applied to the first waves of critically ill covid patients undergoing mechanical ventilation, and to the best of our knowledge none from Italy, one of the countries hardest hit by the pandemic during the early waves.
If the study has an NCT number, this should be shown.
We are thankful to the reviewer for this comment. The study was indeed registered with a NCT number, that we now report in the methods section.
Reviewer 4 Report
Interesting study with some limitations especially the small population of critically ill patients and data collecting of them.
The study can be improved in some ways:
- clearly presented results of AKI Risk score, s-creatinine and s-IL 6 concentrations by the estimated marginal means and standard errors upon ICU admission and 12, 24, and 48 hours after for No/Mild AKI, Moderate AKI and Severe AKI group.
- we need also AUC ROC curve comparing AKI Risk score, s-creatinine concentrations and combination of both parameters for better discriminative performance.
- classification of patients using decision trees including AKI Risk score, s-creatinine concentrations and both parameters at your intervals to measure over all acuracy in percentage (prediction).
- SI units for biochemistry tests are recomended for a log time ago.
Author Response
Reviewer 4
Interesting study with some limitations especially the small population of critically ill patients and data collecting of them.
We appreciate the reviewer's constructive feedback on our manuscript.
The study can be improved in some ways:
- clearly presented results of AKI Risk score, s-creatinine and s-IL 6 concentrations by the estimated marginal means and standard errors upon ICU admission and 12, 24, and 48 hours after for No/Mild AKI, Moderate AKI and Severe AKI group.
We thank the reviewer for this comment, we have now added the requested data to supplemental material. Please note that while we have Aki Risk score at ICU admission, 12, 24 and 48h, we only have IL 6 and creatinine values in emergency department, ICU admission and at maximum values.
- we need also AUC ROC curve comparing AKI Risk score, s-creatinine concentrations and combination of both parameters for better discriminative performance.
We thank the reviewer for this comment, we now add the requested curves in supplemental material 4 with the AUC for each of the three curves.
- classification of patients using decision trees including AKI Risk score, s-creatinine concentrations and both parameters at your intervals to measure overall acuracy in percentage (prediction).
We agree with the reviewer it would have been interesting to apply decision trees to a similar setting. Unfortunately, the small number of participants in the sample, which we now better acknowledge in the limitations sections, prevents the possibility of using decision trees, which would require a considerably larger sample size to be stable and interpretable. We now more accurately identify the limit of the sample's numerosity in discussion and limitation section.
- SI units for biochemistry tests are recomended for a log time ago.
We thank the reviewer for this comment, we have now converted the units of lab values to SI unit, and updated table 2 accordingly
Round 2
Reviewer 1 Report
Many thanks to the authors for their responses to my comments. They have solved most of the doubts, deficiencies and errors detected in the first review.
However, some more modifications must be performed for the publication of the paper:
- TIMP2*IGFBP7 product must be considered as a single marker, so references to it must be made in the singular (“biomarker”, and not “biomarkers”).
- Regarding the secondary outcome: Since objectives must contain an action, the writing seems incomplete (Suggestion: “Secondary outcomes were to analize development”
- Abbreviations must be checked again ( example: the definition of "PEEP" does not appear in the text).
- Figure 1: hardly visible legends, units, etc.; the group legend is not neccesary (it is on the horizontal axis). On the other hand, the units have not been clarified as suggested in the first review What does “val” mean? What do the asterisks mean?
- Figure 2: the scale is not visible. The groups have not been ordered.
Furthermore, the purpose of measuring this marker has not been discussed/explained, as it was already suggested in the first review.
Reviewer 2 Report
The response of the authors is satisfactory to both my comments and the comments of the other peer-reviewers. I believe the paper can be now accepted for publication.
Author Response
Reviewer #2:
The response of the authors is satisfactory to both my comments and the comments of the other peer-reviewers. I believe the paper can be now accepted for publication.
Response: We are grateful to the reviewer for the kind comment.
Reviewer 3 Report
The authors have rewritten and improved several parts of the manuscript, which I now find acceptable for publication.
Author Response
Reviewer #3:
The authors have rewritten and improved several parts of the manuscript, which I now find acceptable for publication.
Response: We are grateful to the reviewer for the kind comment.